

# Sensitivity of the WRF-Chem (V3.6.1) model to different dust emission parametrisation: Assessment in the broader Mediterranean region

Emmanouil Flaounas[1], Vassiliki Kotroni[1], Konstantinos Lagouvardos[1], Martina Klose[2], Cyrille Flamant[3], and
Theodore M. Giannaros[1]

1 National Observatory of Athens, Athens, Greece
2 USDA-ARS Jornada Experimental Range, Las Cruces, NM, USA
3 LATMOS/IPSL, UPMC Univ. Paris 06, Sorbonne Universités, UVSQ, CNRS, Paris, France

Correspondence to: Emmanouil Flaounas (flaounas@noa.gr)

## Abstract

In this study we aim to assess the WRF-Chem model capacity to reproduce dust transport over the eastern Mediterranean. For this reason, we compare the model aerosol optical depth (AOD) outputs to observations, focusing on three key regions: North Africa, the Arabian Peninsula and the eastern Mediterranean. Three sets of four simulations have been performed for the six-month period of spring and summer 2011. Each simulation set uses a different dust emission parametrisation and for each parametrisation, the dust emissions are multiplied with various coefficients in order to tune the model performance. Our assessment approach is performed across different spatial and temporal scales using AOD observations from satellites and ground-based stations, as well as from airborne measurements of aerosol extinction coefficients over the Sahara.

Assessment over the entire domain and simulation period shows that the model presents temporal and spatial variability similar to observed AODs, regardless of the applied dust emission parametrisation. On the other hand, when focusing on specific regions, the model skill varies significantly. Tuning the model performance by applying a coefficient to dust emissions may reduce the model AOD bias over a region, but may increase it in other regions. In particular, the model was shown to realistically reproduce the major dust transport events over the eastern Mediterranean, but failed to capture the regional background AOD. Further comparison of the model simulations to airborne measurements of vertical profiles of extinction coefficients over North Africa suggests that the model realistically reproduces the total atmospheric column AOD. Finally, we discuss the model results in two sensitivity tests, where we included finer dust mode (less than 1 μm) and changed accordingly the dust bins mass fraction.

## 1 Introduction

The geographical belt composed by North Africa and the Arabian Peninsula constitutes the largest desert in the world (Tsvetsinskaya et al, 2002). This region is a major dust source, emitting annually large loads into the atmosphere and thus has a global impact on climate and air quality (Huneeus et al, 2011). While both North Africa and the Arabian Peninsula emit remarkable amounts of particulate matter, it is the Saharan desert that constitutes the worldwide main source of dust. In fact, annual dust emission from the Arabian Peninsula are about one fifth of those from North Africa (Taichu et al, 2006). Dryan et al. (1991) showed that dust intrusions in the Eastern Mediterranean from the Arabian Peninsula have a short duration (of the order of a day) and take place within shallow atmospheric layers of up to 2 km above sea level, while African dust intrusions persist longer (2-4 days of duration) and transport takes place at atmospheric layers over 3 km of altitude.

Climatologically, the emissions of dust over both regions are higher during spring and summer (Engelstaedter et al, 2006; Taichu et al, 2006). During this period, the atmospheric dynamics over North Africa, the Middle East and the eastern Mediterranean are strongly impacted by the monsoon system of West Africa and India (Flaounas et al, 2012; Tyrlis et al, 2014). The Indian monsoon onset establishes a low pressure system that extends from the Indian Subcontinent to the eastern Mediterranean. A quasi-constant descending cell of air masses is located over the eastern Mediterranean with pronounced impact on the surface wind circulation over the region (Tyrlis et al, 2014). Under these conditions dust storms are frequent over the Arabian Peninsula (Miller et al, 2008), while the





Mediterranean climate and dust emissions are strongly affected by the West African monsoon and the Saharan heat low (Chauvin et al, 2010; Wang et al., 2015). Indeed, early summer is of particular interest for West African dust emissions. At the end of June, the monsoon propagates towards the north, displacing the ITD (Intertropical
discontinuity, a near surface convergence zone between the monsoon and the Harmattan wind) to 20°N over the main source areas of dust (Prospero et al. 2002; Sultan et al., 2007; Klose et al., 2010; Gazeaux et al., 2011). Indeed, this region has been estimated by Evan et al. (2015) to emit about 80% of the total North African dust. In particular, Engelstaedter et al. (2006) showed that the West African monsoon onset plays a key role in the regional seasonal maximum of dust emissions. While uptakes of dust may occur due to local meteorological
events such as dust devils, wind surges, turbulent mixing of low-level jets and cold pools associated with convective systems (Bou Karam et al., 2008; Knippertz and Todd, 2012; Klose and Shao, 2013), synoptic scale systems may transport dust away from the continent with a global impact (D'Almeida, 1986; Prospero, 1996; Moulin et al., 1997; Kaufman et al., 2005; Bristow et al., 2010; Bou Karam et al., 2010; Prospero et al., 2014; Flaounas et al., 2015).

Despite the importance of the African continent as a worldwide major dust source, the quantification of dust emissions is still an open question and strongly relies on numerical modelling. However, modelling dust uptake is a delicate issue, subject to a variety of uncertainties associated with the model's capacity to realistically reproduce the near surface meteorological conditions, the applied dust emission parametrisation, the model's
vertical and horizontal resolutions, as well as the surface-related input datasets, such as erodible areas (e.g. Menut et al, 2007; Haustein et al, 2015; Teixeira et al, 2015; Evan et al., 2015). Indeed, the results of the analysis of an ensemble of 15 models showed that the potential dust emissions of North Africa vary significantly, ranging between 400 to 2200 Tg per year (Huneeus et al., 2011).

Accurate forecasts of dust emission and transport are also a societal demand worldwide as they pertain to many health and economic issues, such as air quality. Ambient air pollution is now the world's largest single environmental health risk, causing 3.7 million premature deaths worldwide every year (World Health Organization, 2014; Lelieveld et al., 2015). Modelling dust uptake and transport requires adequate parametrisations, input fields and tuning techniques in order for results to best match observations (Basart et al.,
2012; Benedetti et al., 2014; Sessions et al., 2015). For instance, Flaounas et al., (2009) showed that the realistic simulation of a pollution episode in southern France depended strongly on the explicitly resolved dust emissions over North Africa. In another case study of a three-day dust event over the Bodélé depression in North Africa, Todd et al. (2008) showed that the simulated dust-related fields (such as dust flux and concentration) from five models differed by at least an order of magnitude. The meteorological conditions were realistically reproduced
by all five models, suggesting that uncertainties were mostly related to the dust emission parametrisations and/or corresponding land-surface input data.

In this study, we test the sensitivity of the Weather Research and Forecasting model with chemistry (WRF-Chem) Version 3.6.1 (Grell et al., 2005) to the dust emission parametrisation through the comparison of
modelled and observed atmospheric optical depth (AOD) over a large region that includes North Africa, the Arabian Peninsula and the eastern Mediterranean basin. The WRF-Chem model has been previously used to investigate dust storms and dust interactions with atmospheric thermodynamics and radiation (e.g. Zhao et al., 2010; Smoydzin et al., 2012; Kalenderski et al., 2013). In particular, Su and Fung (2015) used WRF-Chem to assess its performance to simulate dust concentrations over East Asia using two different dust emission
parametrisations. Their results showed significant differences in the WRF-Chem performance when different dust uptake parametrisations were applied. To the best of the authors' knowledge, this is the first comprehensive study in evaluating the model performance with a focus on dust emissions over the area of North Africa, the Arabian Peninsula, and the eastern Mediterranean. Our study concentrates in the six-month period from spring to summer 2011, when dust transport over the Mediterranean is expected to be high. Summer 2011 was included in
the evaluation period in order to benefit from aircraft measurements of aerosol extinction coefficient profiles that were acquired over the Sahara during the Fennec campaign (Ryder et al., 2015).



Our objective is to assess the model performance in key dust source regions. For this purpose, we performed three sets of simulations with each set using a different dust emission scheme. For every dust emission scheme we applied different tuning coefficients to the surface dust emission fluxes (a total of 12 simulations). Model outputs have been compared to AOD, as observed by satellites, ground-based AERONET stations and to airborne lidar-derived extinction coefficient measurements. Retaining the dust schemes in their original configuration, but multiplying dust emissions by different coefficients, is a straight forward tuning of the model performance, in order to achieve realistic AOD values within the simulation domain. However, tuning dust emissions is a secondary objective here that aims to establish an empirically modified model setup that effectively reproduces dust transport over the eastern Mediterranean. The limitation of this approach is that the tuning has no physical basis and hence the model adjustment is only valid for the specific simulation area and model setup (see also Section 5).

## 2 Simulation set-up, observations and methods

### 2.1 Model domain and configuration

The WRF-Chem model was operated on the domain shown in Fig. 1 at a standard longitude-latitude projection with a horizontal resolution of 0.22° and 0.19°, respectively (of the order of ~22km). The domain is composed by 424×250 grid points and 40 vertical levels. All simulations have been performed for the period of 21 February 2011 to 31 August 2011. The model is initialized with zero dust concentrations. The one-week period from February 21 to February 28 has been used as a spin-up period for building dust concentrations within the domain and has not been taken into account for the model assessment. The model was forced into its initial and boundary conditions by the ERA-Interim (ERA-I) reanalysis of the European Center for Medium-Range Weather Forecasts (Dee et al., 2011). Boundary conditions and sea surface temperature were updated every six hours.

The WRF model has been previously shown to realistically simulate the West African monsoon and heat low dynamics during spring and summer (Flaounas et al., 2011; Klein et al., 2015). Here, we use the Grell 3D ensemble scheme for convection (Grell and Devenyi, 2002), the WRF single moment five microphysics scheme (Hong et al., 2004) and the Yonsei University planetary boundary layer parametrization (Hong et al., 2006). In this study we nudged wind, temperature and water vapour at each grid point to the ERA-I reanalysis, except within the boundary layer. Grid nudging has been previously shown to contribute to the realistic reproduction of a severe dust event over India (Kumar et al, 2014), as well as to the atmospheric circulation in seasonal simulations (Lo et al, 2008). The grid nudging coefficient we used is $6\times10^{-4}$ s$^{-1}$. Comparison of 10-meter wind speed between our nudged simulations, a simulation where no nudging was applied and SYNOP (surface synoptic) observations showed that nudging clearly improves the model 10-m wind speed. In particular, it was found that applying nudging reduces the model 10-meter wind speed absolute bias over North Africa by approximately 35%, while it also allows for a better subjective agreement between the observed and modelled synoptic scale patterns associated with dust transport. Furthermore, in long simulations of more than a few days, nudging is beneficial in reducing uncertainties in the atmospheric circulation due to the model internal variability. Our choice to nudge is thus based on achieving realistic seasonal atmospheric circulation over the domain which is particularly important to dust emissions. Since nudging introduces additional tendencies to the model for wind, temperature and water vapour, it would affect our results only if we compared simulations that treat dust direct and indirect effects. However, here dust is treated as a passive tracer.

### 2.2 Model chemistry component and sensitivity tests

The chemistry component of the WRF model is used in dust-only mode, where the model takes into account dust uptakes from the soil -the only source of particulate matter-, and transports it as a passive tracer within the simulation domain, treating explicitly gravitational settling, and vertical mixing. Consequently, all simulations present identical meteorological conditions and atmospheric circulations, i.e. unaffected by dust direct or indirect effects. Three dust emission schemes are considered which output dust emissions for five size bins with effective radii of 0.73, 1.2, 2.4, 4.8 and 8 μm. In all schemes, the areas where dust can potentially be emitted are defined by the erodibility field, used here as a spatial dataset of 1°x1° resolution in longitude and latitude. This field was



defined by Ginoux et al. (2001; Figure 1) aiming to account for the variable amounts of sediment available in
topographic depressions. The erodibility field has values between 0 and 1, expressing the probability of
accumulated sediments to lie in a given location. The following dust emission schemes have been tested:

(a) The first scheme is based on an empirical formulation developed by Gillette and Passi (1988) and is
incorporated in WRF-Chem within the GOCART model (Ginoux et al.,2001). In this scheme (GOCART in the
following), the dust mass flux from the surface to the first model atmospheric level scales with the third power
of 10-meter wind speed, multiplied by the surface erodibility (as defined by Ginoux et al., 2001) and the mass
fraction of each size class. In accordance with Ginoux et al., (2001), mass fraction is set to be equal to 0.1 for
emitted dust of effective radius 0.73 μm, suggesting that clay (particle size smaller than 1μm) corresponds to
10% of the total silt mass. For the other four bins, considered to be silt (effective radii larger than 1μm), it is
assumed that the mass fractions are equally distributed and are 0.25 each. Dust emissions are activated as soon
as 10-meter wind speed exceeds a threshold value. This threshold is calculated for dry soil based on a
formulation derived by Marticorena and Bergametti, (1995) and is then adjusted according to soil moisture.

(b) The second dust emission scheme is the parametrisation developed by Marticorena and Bergametti (1995),
incorporated in WRF-Chem in the Air Force Weather Agency (AFWA) dust module (AFWA hereafter). The
scheme parametrises dust emission caused by saltation bombardment and the vertical dust emission flux is
proportional to the horizontal saltation flux, calculated when friction velocity exceeds a threshold. For dry soil,
the threshold is the same as that used in GOCART emission scheme, but a different soil moisture correction is
used in the AFWA scheme. The horizontal saltation flux is obtained using a modification of the expression
proposed by White (1979). The proportionality between dust emission and saltation flux was empirically related
to soil clay content by Marticorena and Bergametti (1995). Dust emissions in AFWA are also scaled with the
erodibility field from Ginoux et al., (2001) and distributed to the bins according to their mass fraction, derived
by Kok (2011).

(c) The third emission scheme is that developed by Shao (2004) implemented in the University of Cologne (UoC
hereafter) dust module package. The scheme accounts for the emission mechanisms of saltation bombardment
and aggregate disintegration and relates dust emission to the volume removal by saltating particles. In the
scheme of Shao (2004), vertical dust emission flux is also proportional to horizontal saltation flux, but the
proportionality depends on soil texture and soil plastic pressure. The scheme of Shao (2004) was originally
implemented using four particle-size bins, but was modified to have size bins consistent with those used in the
other parametrisations. Required land-surface input data sets for the scheme are soil type and vegetation cover.
In contrast to GOCART and AFWA, the UoC scheme uses the erodible area by Ginoux et al., (2001) only to
define areas of potential dust emission, i.e. dust emissions are calculated only at grid points where erodibility is
non-zero. In contrast to the other two schemes tested in this study, the calculated dust emissions are not scaled
with the erodibility function.

For each dust emission scheme, we perform four simulations where the dust emissions are multiplied by four
different coefficients in order to increase or decrease the dust fluxes in the atmosphere. Preliminary tests showed
that a coefficient equal to 1 for AFWA and GOCART resulted in disproportionly high AOD values over North
Africa compared to the scheme of UoC. Consequently, we chose coefficients to be different for the four
simulations when using the UoC scheme. Table 1 presents a summary of the 12 performed simulations set-up.

### 2.3 Observations and comparison approach

To compare modeled AOD with observations, we use the MODIS AOD observations at 550 nm from the Terra
and Aqua satellites, corresponding to version 6 of daily gridded data in 1°x1° grid spacing in longitude and
latitude, provided by the Goddard Earth Sciences Data and Information Services Center (MOD08 D3 and
MYD08 D3, combined dark target and Deep Blue, giovanni.sci.gsfc.nasa.gov/giovanni). Aqua and Terra
satellites provide worldwide daily observations, having a 2330 km swath and crossing the equator at 1:30 pm
and 10:30 am local time, respectively. Retrieval of MODIS aerosol data is performed by different algorithms





(e.g. Hsu et al., 2004; Remer et al., 2005) according to the underlying surface type. The accuracy of the AOD retrievals has been evaluated both on a global and regional scale, against AERONET sun photometer measurements (e.g. Levy et al., 2010; Sayer et al., 2013). From the MODIS database, we have also used measurements of Ångström Exponent (AE) over land (470−660 nm; MOD/MYD 08_D3_051), over ocean (550−865 nm; MOD/MYD 08_D3_051) and over deserts (412-470 nm; MOD/MYD 08_D3_6), as well as the

absorption Aerosol Index (AI), taken from OMI-Aura (Ozone Monitoring Instrument) measurements (Torres et al., 2007). Following the same approach as in Flaounas et al. (2015) the MODIS AOD dataset was filtered so that model evaluation is performed only for grid points and days that dust is present. For this reason, we took into account only AOD values when AE is lower than 0.7 and AI is greater than 1. Finally, we also use ground observations of AOD, taken by the aerosol robotic network (AERONET, Holben et al, 1998). In contrast to

satellite observations, AERONET observations offer the advantage of continuous, high-temporal resolution measurements in the daytime over a given location where satellite coverage might not be always available.

Figure 1 shows the dust fraction of erodible surface at each grid point, as taken into account by the WRF-Chem simulations. As expected, the major sources of dust are located in North Africa and in the Arabian Peninsula. We

focus on these regions in order to validate the modelled dust emissions. A second focus is on the eastern Mediterranean in order to validate the model capacity in realistically reproducing the dust transport over this region. These three subregions are depicted by boxes in Fig. 1. Six AERONET stations have been chosen so that their locations are representative of the sub-regions of interest and their observations are available during the simulation period (Fig. 1).


Finally, airborne measurements of the lidar-derived extinction coefficient acquired over the western Sahara during the Fennec campaign are used to evaluate the vertical profiles of modelled dust. During Fennec campaign, the SAFIRE (Service des Avions Français Instrumentés pour la Recherche en Environnement) Falcon 20 was equipped with the LEANDRE Nouvelle Génération (LNG) backscatter lidar (Bruneau et al., 2015). The

profiles of atmospheric extinction coefficient at 532 nm were retrieved using a standard lidar inversion method that employs a backscatter-to-extinction ratio of 0.0205 sr −1 (see Schepanski et al., 2013, for details). At this wavelength, the lidar signal is mostly sensitive to aerosols with radii ranging from 0.1 to 5μm, and hence to dust aerosols. Furthermore, over the African continent, close to the sources, desert dust particles are generally considered to be hydrophobic (e.g. Fan et al., 2004). Therefore, extinction associated with desert dust is

generally considered to be a good proxy for dust concentration in the atmosphere. The retrievals have an estimated uncertainty of 15%, a resolution of 2 km in the horizontal and 15 m in the vertical. Lidar-derived extinction coefficient profiles were averaged over 30 min (~350 km) along levelled legs performed by the Falcon 20 during 5 flights on 14, 15, 20, 21 and 22 June (see Ryder et al., 2015 for flight tracks). This was done to extract the main characteristics of the dust layers over the Sahara (vertical extent, magnitude of extinction) in an

integrative approach more adapted to a comparison with model outputs which generally do not reproduce the high-spatial variability observed with lidars. The locations of the averaged vertical profiles are shown as black dots in Fig. 1. The lidar-derived extinction coefficient profiles are compared to their simulated counterparts averaged over the same leg and extracted at the model output time closest to the time when the lidar profiles were acquired.


### 3 Comparison of simulation results to observations

### 3.1 Model assessment in the simulation domain
The seasons of spring and summer are expected to have the highest dust emission activity in the broader region

including North Africa, Middle East and the Mediterranean (Moulin et al., 1998). Figure 2 shows the average dust AOD as retrieved by MODIS for the whole six month period of spring and summer 2011. Over North Africa, the higher AOD values are observed along the 15°N latitudinal belt, at the climatological location of the inter-tropical discontinuity frontal area between the monsoon and the Harmattan wind. The higher AODs are observed downstream of the Bodélé depression. High dust concentrations are also located over the northern part

of the Arabian Peninsula and are related to the Shamal winds continuously blowing over dust sources, linked to



the alluvial plains of Syria, Irak and western Iran. Large AOD values are also observed to be associated with emissions from the Aral Sea sediment basin, east of the Caspian Sea. The mean AOD values of spring and summer are dramatically lower over the Mediterranean region where dust sources are limited.

The AOD differences between the WRF-Chem simulations and the MODIS estimations (as shown in Fig. 2) are presented in Fig. 3. To be consistent with the equator crossing time difference between Aqua and Terra, we compare AOD from MODIS with model outputs at 12:00 UTC. Differences correspond to the AOD six-month averages, taking into account only the days and grid points when MODIS provides measurements. As expected, in all simulations, the AOD bias varies over the whole region as a function of the dust flux coefficient. In
Sim_GOCART-1 and Sim_GOCART-0.75 (Figs 3a and 3d), the AOD is largely overestimated over North Africa while when applying a coefficient of 0.5, the model seems to be in better agreement with the MODIS observations (Fig. 3g). On the other hand, the modelled AOD over the Mediterranean Sea seems to be closer to the observations in Sim_GOCART-1 and Sim_GOCART-0.75, while the model overestimates AOD over North Africa. In the Arabian Peninsula, Sim_GOCART-1 tends to overestimate AOD over the southeastern part of the
region, compared to the AOD over the northern side. This is consistent with the higher fraction of erodible surface in the south of the Arabian Peninsula, as shown in Fig. 1. Sim_GOCART-0.75 also appears to produce the most realistic AODs in that region. It is noteworthy that all the GOCART simulations underestimate the AOD in the vicinity of the Euphrates and Tigris rivers basin.

Similar results are obtained using the AFWA and UoC schemes. Over North Africa, the AOD is overestimated for the larger tuning coefficients (Figs 3b and 3c), with a smaller bias over the Mediterranean. Sim_UoC-1.5 appears to show the smallest bias of all UoC simulations over North Africa. In fact, the UoC simulations tend to present a large overestimation of AOD over three hot spots of dust emissions located in southern Iran, close to the Sistan region, in the northern part of the horn of Africa and in the eastern part of central Africa. These are
areas which have a small fraction of erodible surface (compare Fig. 1), thus emissions produced with the GOCART and AFWA schemes are already significantly reduced through multiplication with this fraction (Section 2.1). It cannot be ruled out that similar overestimations would occur without this second scaling. Overall, the lower tuning coefficients provide a general underestimation of AOD over the whole simulation domain, regardless the dust emission parametrization (Figs 3j, 3k 3l).

In order to quantify the WRF-Chem model skill in reproducing the six-month average AOD in all simulations, Fig. 4 presents the spatial Taylor diagram (Taylor, 2001) that compares MODIS observations (as presented in Fig. 2) to the simulation outputs. In the Taylor diagrams, the centred root mean square error (RMSE, in abscissa) provides a measure of the model total AOD differences from the observations within the entire domain, while the
standard deviation (in ordinate) and correlation provide a measure of the models skill to reproduce the AOD spatial variability. Figure 4 shows that simulations using GOCART and AFWA present a correlation coefficient of the order of 0.5, while in simulations using UoC correlation coefficient is about 0.3. This suggests a rather similar skill of GOCART and AFWA simulations in reproducing the seasonal spatial variability of dust concentrations. On the other hand, the RMSEs and standard deviations strongly depend on the applied tuning
coefficients. In fact, Sim_GOCART-0.25, Sim_AFWA-0.25 and Sim_UoC-0.5 seem to present standard deviations which are closer to MODIS, as well as the lowest RMSE. Although these three simulations underestimate the AOD compared to MODIS (see Figs 3j, 3k and 3l), their overall bias -averaged over the whole domain is smaller than in the simulations using larger tuning coefficients as for instance Sim_AFWA-0.75, Sim_GOCART-0.75 and Sim_UoC-1.5 (Fig. 3d, e, f, respectively). Small and moderate tuning coefficients limit
the simulated hot-spots of high dust concentrations and thus the model standard deviation is closer to the observations. When comparing the simulations in their standard set-up (tuning coefficient equals to 1), the UoC scheme shows the smallest standard deviation and RMSE, followed by AFWA and GOCART.

### 3.2 Model assessment on regional scale
In order to evaluate the model skill in reproducing the AOD on regional scales, we focus on three sub-domains, outlined in Fig. 1a. For each simulation, Figure 5 shows the average absolute bias of modelled AOD with respect



to MODIS derived AOD within each sub-domain and for the whole six-month simulation period. For each dust emission parametrisation, there is a coefficient that corresponds to a minimum absolute bias. As discussed in the previous section, all three simulation sets provide smaller biases over the Eastern Mediterranean domain when the tuning coefficients are large (Fig. 5c). On the other hand, smaller tuning coefficients seem to be more adequate for the North African domain. Indeed, Sim_GOCART-0.5 and Sim_AFWA-0.5 result in smaller biases for the African domain (Fig. 5a), while Sim_GOCART-1 and Sim_AFWA-1 tend to produce smaller biases for the Eastern Mediterranean. Sim_UoC-1.5 achieves a minimum absolute bias over the North African domain (Fig. 5a), while Sim_UoC-2 yields the minimum absolute bias for the Eastern Mediterranean domain (Fig. 5c). Regardless the dust emission parametrisation, the North African domain and the Arabian Peninsula do not share the same tuning coefficients for minimizing absolute errors despite that they are both regions with major dust sources. Indeed, Fig. 5b shows that larger tuning coefficients in GOCART and AFWA (Sim_GOCART-0.75 and Sim-AFWA-1) tend to reproduce smaller biases in the Arabian Peninsula. There is an opposite behaviour of the simulation results obtained with UoC. In fact, Sim_UoC-1.5 produces a smaller bias for North Africa, while Sim_UoC-1 (no tuning) yields a better performance for the Arabian Peninsula. Such a different behaviour between the schemes might be attributed to the different treatment of the potential dust source areas. Overall, the use of coefficients in order to tune the modelled dust emissions is shown to reduce or increase the model absolute bias of AOD over the chosen regions of interest, namely North Africa, the Arabian Peninsula and the Eastern Mediterranean. The optimal coefficient to minimize the regional AOD absolute bias is not the same for all regions.

To gain further insight on the capacity of WRF-Chem to reproduce the regional AOD, Fig. 6 shows time series of the daily evolution of AOD from WRF-Chem and MODIS, averaged over each of the three domains and Fig. 7 shows Taylor diagrams that statistically assess the model using the time series as shown in Fig. 6. For Africa, both model and MODIS show a strong overall variation in the domain averaged AOD, with few distinct peaks during the investigation period. All simulations qualitatively capture the timing of most periods with increased AOD, however, the double peak in late June and early July is not well reproduced by the parametrisations. In fact, Sims_AFWA show slightly better correlations compared to Sims_GOCART and Sims_UoC, suggesting that the simulations using AFWA applies better to North Africa for the given model set-up (i.e. for the given domain, resolution etc.). A similar result is obtained for the Arabian peninsula domain shown in Fig. 6b, except that Sims-UoC strongly overestimate dust emissions starting from July onward. Correlation coefficients are also slightly higher for Sims_AFWA for the Arabian Peninsula than for the two other simulation sets.

For the Eastern Mediterranean, Fig. 6c shows that the MODIS AOD observations present an average AOD background value of the order of 0.2, while several peaks are representative of major dust transport events (as for instance on 1 May 2011; Fig. 6c). Since the atmospheric circulation is identical in all simulations and nudged to the ERA-I reanalysis, the model realistically captures the time of the dust transport events, as reflected by the high correlation coefficient of about 0.7 for all simulations (Fig. 7c). On the other hand, if no dust transport takes place (as for instance during the second half of June 2011 in Fig. 6c) the WRF-Chem model AOD values are close to zero (Fig. 6c). Consequently, regardless the dust emission scheme, the model fails to realistically reproduce the background dust concentration over the Mediterranean. It is thus plausible to suggest that if no major dust transport event takes place in the region, the model excessively removes dust from the atmosphere over the Mediterranean and/or that other aerosol sources are not captured by the model. Given that our motivation is to assess the WRF-Chem eperformance especially in reproducing dust transport over the Eastern Mediterranean, in Fig. 8 we provide an example of the model performance in simulating a dust episode that took place in July 23 of 2011. The model is compared to AOD from the AERUS-GEO product which only had few missing values for this date compared to MODIS. AERUS-GEO is derived by observations of the Meteosat Second Generation Spinning Enhanced Visible and Infra-Red Imager (MSG/SEVIRI; Carrer et al., 2014). Observations clearly show high AOD values, ranging from 0.5 to 1, that extend from the African coast towards the central Mediterranean Sea, between Sicily and Greece (Fig. 8a). Figures 8b, 8c and 8d show the simulations performance using a tuning coefficient of 1 which yielded better results for the Eastern Mediterranean (Fig. 5). The modeled AODs vary between the simulations, with the GOCART and the AFWA schemes yielding higher





values compared to the UoC scheme. Both GOCART and AFWA simulations seem to produce similar spatial patterns of the dust transport episode and AODs. Since meteorology is identical to all three simulations, the similarity is caused by the AFWA and GOCART emission schemes. Indeed, the same tuning coefficients lead to a similar AOD bias (e.g. Fig. 3) and fairly close correlation coefficients (e.g. Figs 4 and 7). A plausible explanation could reside in the fact that both schemes share the same parametrisation for dry soil threshold friction velocity and that both simulations use soil erodibility to scale dust emission fluxes. Despite their differences, all simulations successfully captured the dust transport event, as a meso-scale tongue of high AOD values. In the next section we focus on smaller scales in order to assess the model performance in reproducing finer spatial features of dust events.

### 3.3 Model assessment at the local scale

AOD observations acquired from AERONET stations allow to assess the skills of WRF-Chem in reproducing the AOD on local scale. Figure 9 shows the model simulated time series of AOD, interpolated at the locations of the six AERONET stations shown in Fig. 1. For the statistical assessment of WRF-Chem at the locations of the AERONET stations, we show the Taylor diagrams corresponding to the time series of Fig. 9, in Fig. 10. In North Africa, all simulations capture the increase of AOD in Zouerate after 15 June (Fig. 9a), i.e. during the period of installation of the Saharan heat low (Todd et al., 2013) over the central Sahara after the African monsoon onset took place (Cornforth et al., 2012). All simulations show equal correlation coefficients of about 0.7 regardless of the tuning coefficients applied to the dust emissions (Fig. 10a). In agreement with the model results over the entire North African domain (Figs 6a and 7a), simulations with tuning coefficients smaller than 1 tend to result in smaller RMSEs and standard deviations which are close to the observations. The modelled AODs at Tamanrasset and Oujda are also in good agreement with the AERONET observations (Fig. 9b and 9c). While at Oujda all simulations present equal correlation coefficients (Fig. 10c) as in Zouerate (but with a correlation of 0.4), the correlations at Tamanrasset depend on the dust emission parameterization (Fig. 10b). This is due to the fact that in Tamanrasset, dust-related AODs depend on both long-range transport from remote North and East Africa sources and local emissions (Cuesta et al., 2008). Simulations using GOCART show larger correlations than the simulations using AFWA and UoC, suggesting a more realistic daily variability of dust concentrations over this site.

At the Solar Village in the Arabian Peninsula, larger correlation coefficients (~0.6) are obtained for the simulations using GOCART. All simulations tend to underestimate the standard deviation and have RMSEs of more than 0.3 (Fig. 10d). Indeed, all simulations seem to underestimate the average AOD during all the six-month period (Fig. 9d). In consistency with the model results at Tamanrasset, the GOCART simulations at the Solar Village present better correlations than both AFWA and UoC. It is rather difficult to explain the reasons for this consistency in the model performance. However, it seems that in both cases, convection may be largely connected with dust outbreaks (Guirado et al, 2014; Houssos et al, 2015).

For the Mediterranean, we compare the AERONET station observations in Crete and Lampedusa with the model results. All simulations were able to reproduce the major dust transport events corresponding to the peaks in AOD (Fig. 9e and 9f). This is also reflected by correlation coefficients of the order of 0.6 (Crete) and 0.7 (Lampedusa) for all simulations, as shown in Fig. 10f and Fig. 10e, respectively. It is noteworthy that the model assessment in capturing dust transport on local scales is a delicate issue. For instance, the event shown in Fig. 8a seems to affect Lampedusa but has a limited impact on Crete. Indeed, Fig. 9e shows that the AERONET station captures a rise of AOD values during late July, while in Crete there is no such trend in the observations. In contrast, all simulations in Fig. 8 seem to extend the dust transport to more eastern locations (Fig. 8b, 8c and 8d) and hence when compared to AERONET over Crete they overestimate AOD in late July (Fig. 9f). Despite the high correlation coefficients, all simulations underestimate the background dust concentration in these stations. Indeed, all simulations show AOD values close to zero except when dust transport events take place (Figs 9e and 9f). On the other hand, AERONET observations from both Crete and Lampedusa present values close to 0.2, consistent with the MODIS average regional AOD values shown in Fig. 6c.





### 3.4 Model assessment of the vertical distribution of dust over the Sahara

To gain further insight in the model's capacity to reproduce dust uptake and transport, Fig. 11 shows the vertical profiles of the lidar-derived extinction coefficients from five flights between 14 and 22 June 2011, as well as the corresponding values obtained with WRF-Chem. Airborne measurements have been taken over Northern Mauritania and Northern Mali, in the vicinity of major dust sources (Fig. 1). We present only Sim_GOCART-0.5, Sim_AFWA-0.5 and Sim_UoC-1.5 which have the smallest biases over these locations among all
simulations (Fig. 3). Results are highly variable depending on the flight. In Fig. 11a, all models seem to capture the vertical profile shape with a decrease of dust concentrations with increasing height, and a sharp decrease in extinction around 5 km amsl, marking the top of the Saharan atmospheric boundary layer (SABL). Nevertheless, the model seems to overestimate the total AOD due to excessive dust concentration throughout the atmospheric column. The lidar-derived extinction profile acquired in 14 June is representative of the low dust concentration
over Northern Mauritania and Northern Mali when the Western Sahara was under the influence of cold air masses from the Atlantic (Todd et al., 2013). In subsequent flights, lidar profiles were acquired while the Western Sahara was under the influence of the approaching Saharan heat low as well as strong low-level northeasterly wind surges from the Mediterranean (Todd et al., 2013). The wind surges were responsible for enhanced emissions in the Western Sahara and for the large AODs observed in Zouerate during the second half
of June (Fig. 9a). As for Fig. 11a, the observations in Fig. 11b show that dust concentrations tend to decrease with height, large extinction coefficient values being observed near the surface as the result of dust emissions.

The SABL corresponds to a deep layer (~6km agl in summer) which tends to be fully mixed no earlier than around 18:00 local time. In the daytime, the SABL is composed of a convective mixing layer developing within
a residual layer, so that lidar and dropsonde data acquired around mid-day generally exhibit a two-layer structure (Ryder et al., 2015; Chaboureau et al., 2016). Dust concentrations within the lower half part of the SABL are representative of local emissions while the upper part is dominated by dust transport (Chaboureau et al., 2016). During the Saharan heat low phase, i.e. on 15, 20, 21 and 22 June, lidar data evidence essentially a two-layer structure in the SABL, with a deep well-mixed upper layer (above 1-1.5 km amsl, Fig. 11b-f) and a lower
atmospheric layer of enhanced extinction (below 1-1.5 km amsl). The model fails to capture the observed high extinctions in the lower layer, but has a fairly good performance at reproducing the structure of the SABL as well as the magnitude of the extinction coefficients derived from lidar. The extinctions in the upper part of the SABL are associated with the long-range transport of dust from remote north and easterly sources, a process that is well captured by the model. On the other hand, extinctions in the lower layers are related to small scale processes that
are not captured by the simulations owing to the relatively coarse mesh size of the model (i.e. 22 km). A striking example of that is shown in Fig. 11d, where the low-level extinction values were observed to be the largest during the Fennec campaign. They were caused by the cold-pool of a convective system having developed overnight over the Atlas Mountains, which then propagated south-westward over the Sahara (Todd et al., 2013; Ryder et al., 2015; Chaboureau et al., 2016) and was sampled by the lidar. The development of the convective
system over the Atlas and the related cold-pool can only be captured by convection permitting models as shown by Chaboureau et al. (2016) with mesh size on the order of 5 km or less. Except maybe for the 21 June case, under the particular circumstances detailed above, the Sim_GOCART-0.5 simulation always exhibits the largest extinction coefficients in the SABL. For most flights, the simulated extinction profiles were seen to lay within the observed values if we account for the natural variability sampled by lidar along the Falcon legs.


## 4 Discussion

### 4.1 On the relation between dust concentration and AOD

Overall, the results of the model comparison against observations showed that the modelled spatial and temporal
variability in AOD is rather insensitive to the coefficient applied to the dust emissions. In fact, all simulations using the same dust emission scheme tend to present the same correlation coefficients when compared to observations, whether we consider local scales or the entire domain. In terms of the AOD level in the eastern Mediterranean, the model failed to reproduce the regional background value (of the order of 0.2). This was shown over the regional domain in Fig. 6, as well as in the local AERONET stations of Crete and Lampedusa





(Fig. 9). However, the model showed good skill in capturing the dust transport events. Indeed, the modelled
AOD time series over the eastern Mediterranean presented a large correlation coefficient of 0.7, when compared
to the MODIS observations (Fig. 7c). Model comparison with AERONET presents some limitations. While the
model calculates only dust related AOD, the AERONET measurements may be also representative of other
particulate matter (e.g. sea salt). To gain more confidence in that the 0.2 value of AOD background in
AERONET is due to dust, we compared the MODIS AOD retrievals with the AERONET measurements.
MODIS measurements have been filtered using the criteria AE<0.7 and AI>1 in order to be representative of
dust and have been interpolated to the locations of the AERONET stations at Lampedusa and Crete. The AOD
median from MODIS (~0.2) at these locations has been indeed found to be close to the AERONET median.

Our results derived from AOD observations compare reasonably well to model outputs. However, the AOD
reflects the dust load within the atmosphere and provides no information on the vertical distribution of dust
concentration. Figure 12 shows the near-ground average dust concentration (i.e. the dust concentration at the first
model level) during the whole six-month period for each simulation. The three simulations using the larger
coefficients (Figs 12a, 12b and 12c) show average dust concentrations over North Africa which exceed 1200 µg
m$^{-3}$ in some areas. No PM10 observations were available for performing a long-term direct comparison with the
model simulations; however, the near ground modeled dust concentrations seem to be excessively overestimated,
especially for GOCART and AFWA by default simulations (Aim_GOCART-1 and Sim_AFWA-1). Indeed,
PM10 observations along the Sahel (along ~14°N) are typically less than 100 µg m -3 in spring and summer
(Marticorena et al., 2010). In addition, the comparison of the order of magnitude of modeled dust concentrations
with measurements at specific stations in the Mediterranean (Pey et al., 2013), shows that the model tends to
produce dust concentrations that are one order of magnitude larger than observations during episodes of dust
transport over the Mediterranean (not shown). Consequently, relatively small coefficients (such as the ones used
at Sim_GOCART-0.25, Sim_AFWA-0.25 and Sim_UoC-0.5) seem to be more adequate for the proper
representation of dust concentration over the African continent and for dust transport in the Mediterranean. On
the other hand, our results in Fig. 3 suggest that these simulations lack a realistic representation of AOD within
the whole simulation domain. In order to achieve overall realistic values of AOD, the WRF-Chem model
configurations assessed here produce very large dust surface concentrations. Consequently, there is a
counteracting effect on the model's performance between modelled AOD and dust concentration. More realistic
values of AOD would demand unrealistically high dust concentrations and a realistic model reproduction of dust
concentration yields too small AOD values.

On local scale, vertical profiles of extinction coefficients obtained from aircraft measurements can be used as a
proxy for the vertical profile of dust concentration. Our results in Fig. 11 show that even with different dust
emission parametrisations, simulations tend to reproduce similar profiles, i.e. extinction coefficient profiles
decreasing with increasing height. Nevertheless, differences are observed for simulations using different dust
emission parametrisations. For instance in Fig. 11d, the simulation using GOCART slightly overestimates the
dust extinction coefficients above the altitude of 5 km and underestimates them below. AOD is a convenient field
for assessing chemistry transport models since simulations may be compared to observations from a network of
ground stations and satellites. On the other hand, these observations might provide misleading results on the
model performance. Indeed, due to compensating biases in the upper and lower part of the SABL
(overestimation/underestimation of the extinction coefficients above/below 1.5 km), the AOD values derived
from the simulated profiles are found to be realistic, especially for Sim_GOCART-0.5 and Sim_UoC-1.5 during
the second half of June at the time of the so-called Sahara heat low phase (Fig. 11b-e). These compensating
biases were also highlighted by Chaboureau et al. (2016), even for higher resolution simulations performed with
convection permitting models. Here we presented only five profiles of extinction coefficient, but averaged over
several hundreds of kilometres along the flight legs to average observed outliers, which are thought to be
representative of the model performance.

### 4.2 On the model sensitivity to dust bins size and mass fraction



The effective radii of the dust particles considered by WRF-Chem, mostly refer to coarse particles of more than 1µm. However, dust transport over the Mediterranean is also related to smaller particles (Polymenakou et al., 2008). In addition, the dust aerosol extinction efficiency is expected to be maximum for dust particles of sizes around 0.5 µm which are not taken into account by the 5-bin parametrisations in WRF-Chem. To investigate the potential of improving the model performance in reproducing both realistic AOD and near-ground dust

concentrations, we implemented eight dust-size bins in WRF, following Basart et al. (2012). Two additional sensitivity tests have been performed using only the dust emission parametrisation of GOCART and eight dust-size bins with radii 0.15, 0.25, 0.45, 0.78, 1.3, 2.2, 3.8 and 7.1 µm.

        Changing the size and number of the GOCART dust bins also requires to attribute to each dust bin its fraction

from the total emitted dust mass. In consistency with Ginoux et al. (2001) we considered that the first four size bins correspond to clay and hence to the 10% of the total emitted mass of silt. Therefore, in our first sensitivity test (EXP1), we set the mass fraction for 13each of the first four size bins to 0.025 and for the other four size bins (corresponding to silt) to 0.25. Equal mass fractions per bin within the same dust-size class seems, however, to be unrealistic. To address this issue, in our second sensitivity test we applied the distribution function by Kok

(2011), similar to the AFWA parametrisation. Figure 13 presents the mass fraction of the eight size bins for EXP1 and EXP2, as well as for Sims_GOCART. For all simulations a tuning coefficient of 0.5 has been applied. In order to assess the model sensitivity to changes in the number, radii and mass fraction of the dust bins, our results from the additional sensitivity tests are compared to Sim_GOCART-0.5.

Figure 14 shows EXP1 and EXP2 average difference in near ground dust concentration from Sim_GOCART-0.5 (Figs 14a and 14b), as well as their AOD difference from MODIS (Figs 14c and 14d). Results show that dust concentrations in EXP1 are overestimated all over the dust source areas, with respect to Sim_GOCART-0.5. On the other hand, EXP2 overestimates dust emissions mostly in northwest Africa. It is also noteworthy that in both EXP1 and EXP2, higher dust concentrations are transported to the Mediterranean. Consequently, the AOD bias

between MODIS observations and EXP1 and EXP2 are significantly different than of Sim_GOCART-0.5 (Fig. 3h). However, by repeating a statistical assessment of EXP1 and EXP2 against MODIS observations (as in Fig. 4), we found quasi-equal spatial correlations. Standard deviation and centred RMSE varied according to the simulation (not shown). Regardless the changes in dust bins radii, number and mass fraction, our results for EXP1 and EXP2 seem to have equivalent results as if tuning dust emissions with different coefficient. Here, we

followed the GOCART assumption of equal silt mass fraction but we also took into consideration a more realistic mass fraction distribution from Kok (2011). It would be interesting to adjust the dust bins mass fraction distribution per region, however, this is a rather challenging issue due to the lack of systematic observations over North Africa.

**5 Summary and conclusion**
        In this study, we assessed the WRF-Chem model capacity to realistically reproduce the dust AOD over the broader region of North-Africa, the Middle East and the Mediterranean for the six-month period from spring to summer 2011. We performed three sets of simulations, each using a different dust emission parametrization. For each simulation set we multiplied different tuning coefficients to the parametrized dust emission fluxes, aiming

at minimizing the model's AOD bias in different regions. Our approach resided in comparing the model results to AOD observations across different temporal and spatial scales, using satellite, ground-based and airborne observations.

        The meteorological conditions and atmospheric circulation were identical in all simulations. Therefore, all

differences in AOD originated from the different dust emission parametrisations. When compared to AOD observations, the assessment of the simulations showed that regardless the coefficient used, the model produces similar correlation coefficients for the simulations that use the same dust emission scheme. Consequently, tuning the emissions by a coefficient resulted only in reducing or increasing the AOD model bias. When considering regional or time averaged model outputs, all three different parametrisations that we tested seemed to present

quasi-equal correlation coefficients with the observations. However, when comparing model outputs to local





stations, in four out of six stations (Tamanrraset, Lampedusa, Crete and Solar village), the simulations using GOCART and AFWA presented slightly larger correlation coefficients than UoC. In its default implementation (i.e. using a tuning coefficient of 1), the simulation using UoC showed smaller RMSEs for four out of the six stations than those using GOCART or AFWA. Comparing the model to airplane observations -and given its spatial resolution- the model shows a fairly good skill in reproducing the vertical profiles of extinction coefficients over Northwest Africa. Overall, the GOCART and AFWA simulations presented similar dust emissions with respect to UoC simulations.

The motivation of this study is to determine an adequate model set-up in order to properly reproduce dust concentrations over the eastern Mediterranean. Therefore, a simulation presenting the smallest bias and largest correlation coefficient would be the most adequate choice. However, our results show that there is no optimal model set-up that could minimize bias simultaneously in all three regions of interest. Consequently, the simulations with low coefficients of the order of 0.5 seem to provide a reasonable trade-off choice in order to properly reproduce major dust transport events over the Eastern Mediterranean, as well as realistic levels of AOD over the desert belt of North Africa and the Arabian Peninsula.

Empirical tuning of dust emissions has no physical basis and corresponds to a model adjustment that is valid for the specific model setup (e.g. grid spacing, number of vertical levels, physical parametrisation). In fact, applying tuning only modifies linearly the model performance. Optimization of dust emissions would demand modifications of the parametrization (e.g. change the thresholds of surface and friction wind speeds) or the relevant surface fields (e.g soil erodibility). Such modifications focus on modelling assumptions and thus provide a more physics oriented optimization of the model performance. Given the differences in the physical assumptions of the dust schemes, such sensitivity tests could only focus however on specific parametrisations yielding nonlinear effects on the results.

Future work will be concentrated to further test the model sensitivity to realistically reproduce dust transport events using both eight dust size bins and finer model resolutions. Furthermore, we will concentrate on the climatology of dust transport over the Mediterranean by performing long term simulations also aiming at investigating the aerosols direct and indirect effect.

**Code availability**

The code used in this study corrsponds to the paramterisation of the chemistry component of the WRF model, currently available through the WRF download webpage: http://www2.mmm.ucar.edu/wrf/users/download/get_source.html. The eight bins integration to the model is available upon request to the corresponding author.

**Acknowledgments**

This publication was supported by the European Union Seventh Framework Programme (FP7-REGPOT-2012-2013-1), in the framework of the project BEYOND, under Grant Agreement No. 316210 (BEYOND-Building Capacity for a Centre of Excellence for EO-based monitoring of Natural Disasters). The authors are grateful to NASA for providing the AERONET and MODIS datasets, as well as to the PIs and associated teams for the datasets maintenance and availability. Analyses and visualizations used in this study were produced with the Giovanni online data system, developed and maintained by the NASA GES DISC. Airborne data were obtained during the FENNEC campaign using the Falcon 20 Environment Research Aircraft operated and managed by Service des Avions Français Instrumentés pour la Recherche en Environnement (SAFIRE, www.safire.fr), which is a joint entity of CNRS, Météo-France, and CNES. The Fennec-France project was funded the Agence Nationale de la Recherche (ANR 2010 BLAN 606 01), the Institut National des Sciences de l'Univers (INSU/CNRS) through the LEFE program, the Centre National d'Etudes Spatiales (CNES) through the TOSCA program, and Météo-France. Finally, we are thankful to METEO-FRANCE/CNRM/ICARE for making available the data from MSG/SEVIRI AERUSGEO through the ICARE Data Center (http://www.icare.univ-lille1.fr/).





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

**Tables**

| Simulation names | Description |
|---|---|
| Sim_GOCART- ## | Dust emissions after Ginoux et al. (2001) <br> ## stands for the coefficient multiplying emissions: 1, 0.75, 0.5 and 0.25 |
| Sim_AFWA- ## | Dust emissions based on Marticorena and Bergametti (1995) <br> ## stands for the coefficient multiplying emissions: 1, 0.75, 0.5 and 0.25 |
| Sim_UoC- ## | Dust emissions after Shao (2004) <br> ## stands for the coefficient multiplying emissions: 2, 1.5, 1 and 0.5 |

**Table 1** Simulations description


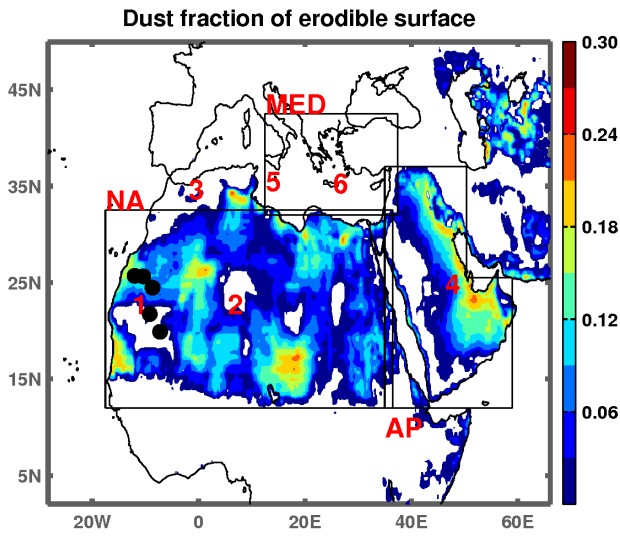

**Figure 1**: Fraction of erodible surface after Ginoux et al. (2001). Boxes depict the three sub-regions of North
Africa (NA), Arabian Peninsula (AP) and the Eastern Mediterranean (MED). Numbers represent the locations of
AERONET stations used in this study and black bullets show the locations of airplane retrievals of the vertical
profiles of extinction coefficients. The AERONET stations are: (1) Zouerate, (2) Tamanrasset, (3) Oujda, (4)
Solar Village, (5) Lampedusa, and (6) Crete.




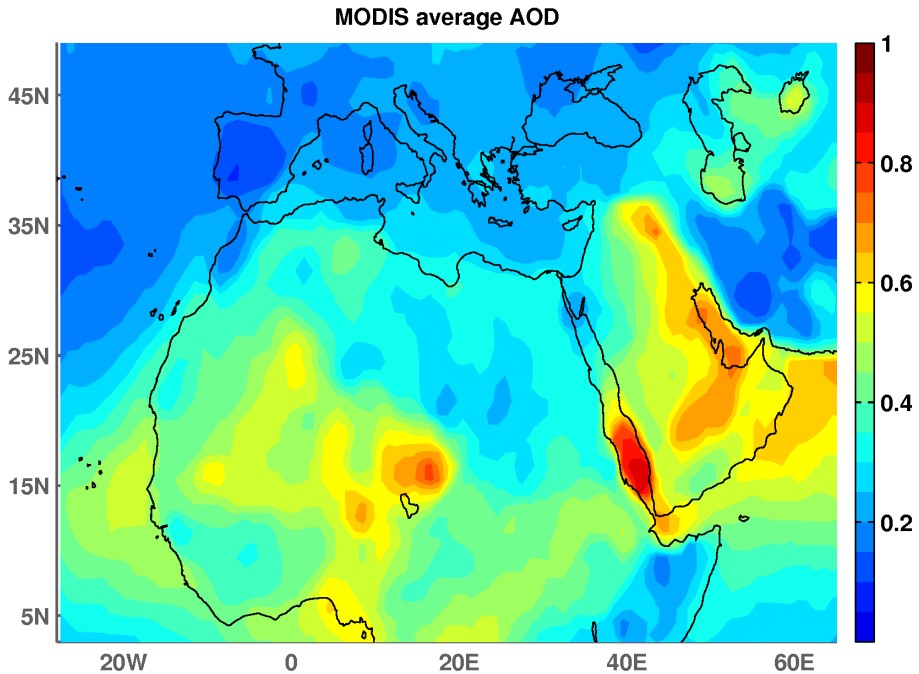


**Figure 2:** MODIS AOD observations within the simulation domain, averaged for the whole six month period, i.e. 1 March to 31 August 2011.





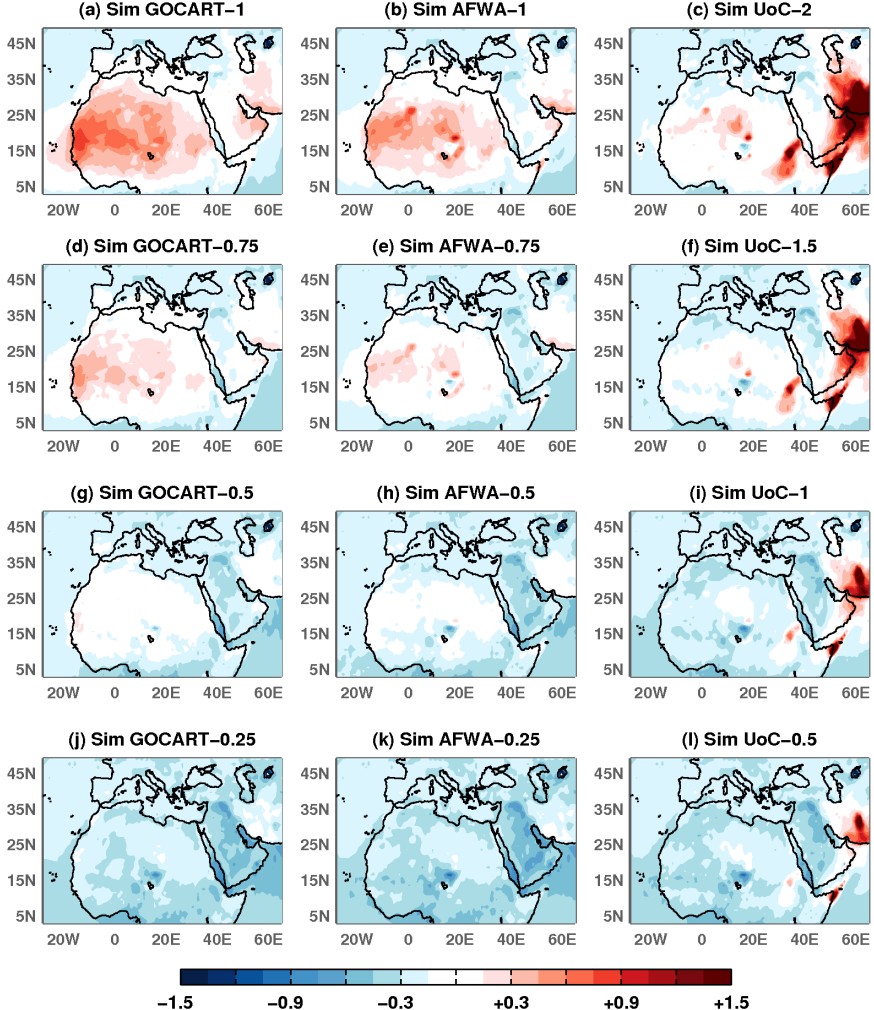

**Figure 3:** Differences between the modeled and observed AOD, averaged over the 6-month period. Note that different coefficients are applied for simulations using UoC compared to the ones using GOCART and AFWA.






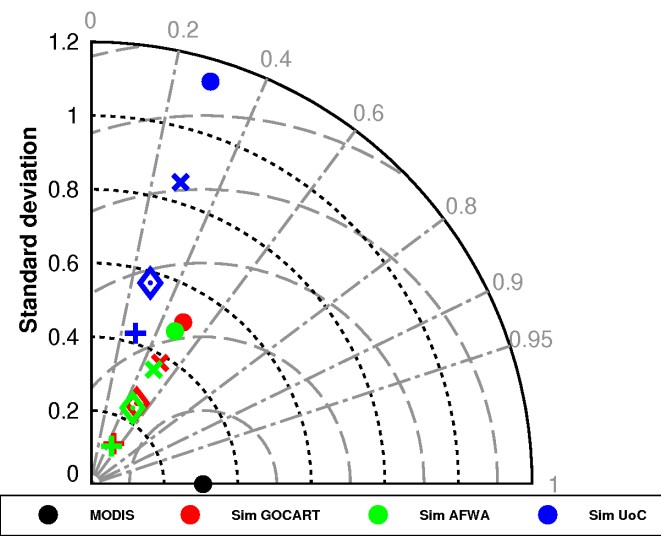

**Figure 4:** Taylor diagram comparing the six-month AOD average of all simulations with MODIS observations for the region illustrated in Fig. 2. Root mean square error lines (gray dashed circular lines) and standard deviations ( blacked dotted lines) are plotted with an interaval of 0.2, while correlation coefficients are shown by the gray radii lines. Symbols in red stand for Sims_GOCART, in green for Sims_AFWA and in blue for Sims_UoC. The black dot stands for MODIS, dots (.) for Sim_GOCART-1, Sim_AFWA-1 and Sim_UoC-2, (X) for Sim_GOCART-0.75, Sim_AFWA-0.75 and Sim_UoC-1.5, Diamond (◊) for Sim_GOCART-0.5, Sim_AFWA-0.5 and Sim_UoC-1, cross (+) for Sim_GOCART-0.25, Sim_AFWA-0.25 and Sim_UoC-0.5.

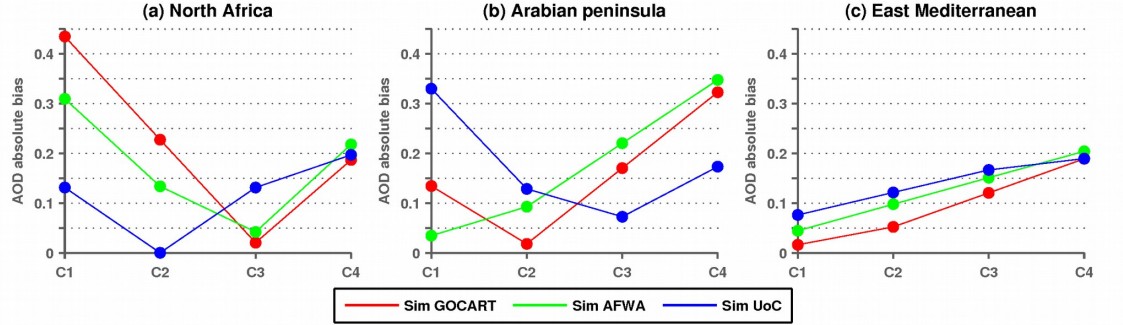

**Figure 5:** Average absolute bias between the simulations and MODIS observations for the whole six-month period and for the three sub-domains, depicted in Fig. 1. The x-axis values of C1, C2, C3 and C4 correspond to the coefficients applied for each simulation set. C1 equals 1, 1 and 2 for Sims_GOCART, Sims_AFWA and Sims_UoC, respectively. C2 equals 0.75, 0.75 and 1.5, C3 equals 0.5, 0.5 and 1 and C4 equals 0.25, 0.25 and 0.5.





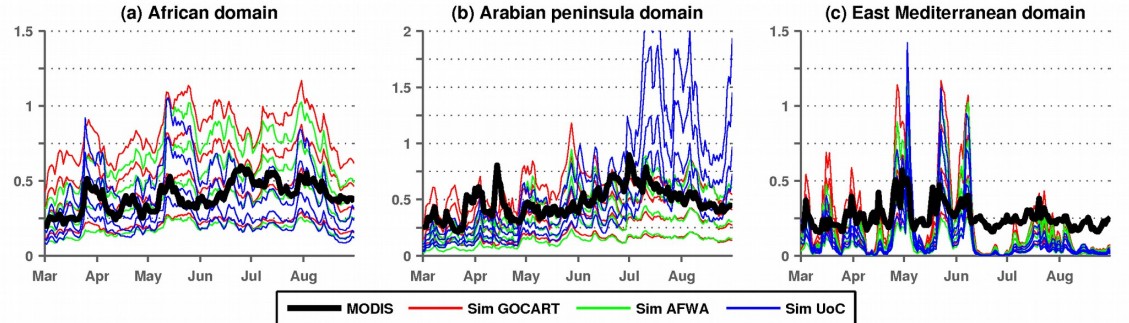

**Figure 6:** Time series of the daily averaged AOD for the simulations and MODIS for the whole six-month period, averaged over the three sub-domains depicted in Fig. 1.


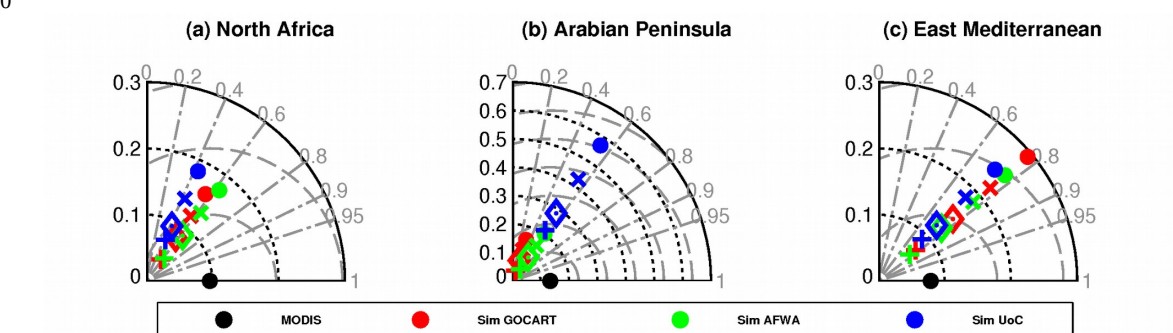

**Figure 7:** Taylor diagram comparing time series of AOD for all simulations to the MODIS observations as shown in Fig. 6. Root mean square error lines are plotted with a 0.1 interval. Symbol annotations are the same as in Fig. 4.






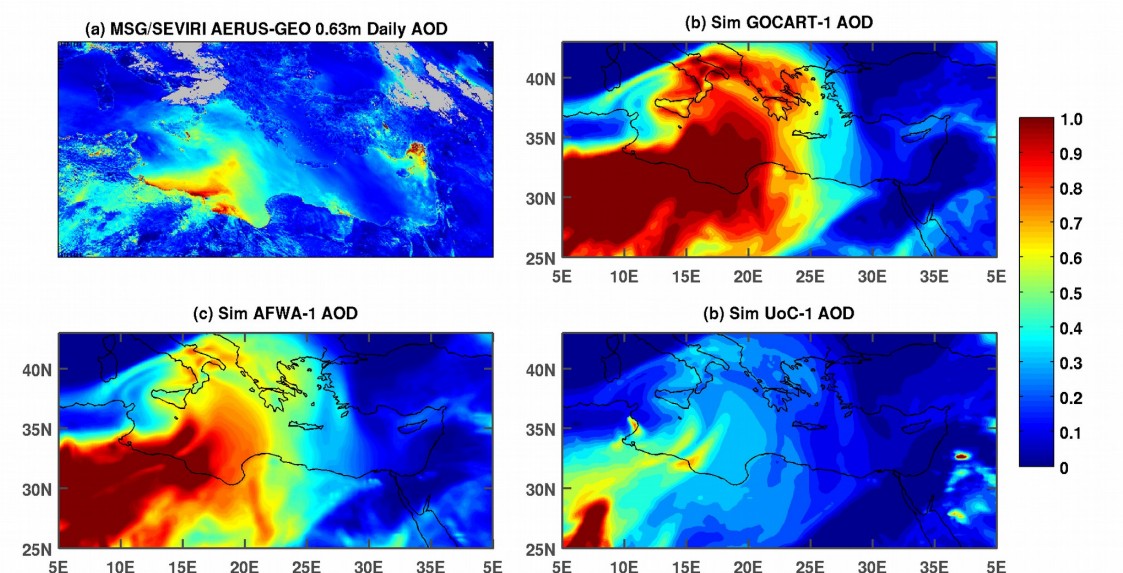

**Figure 8:** AOD over the eastern Mediterranean in July 23 as estimated by the AERUS-GEO product and as simulated by WRF-Chem using default dust emission parametrisations where no tuning coefficients are applied.





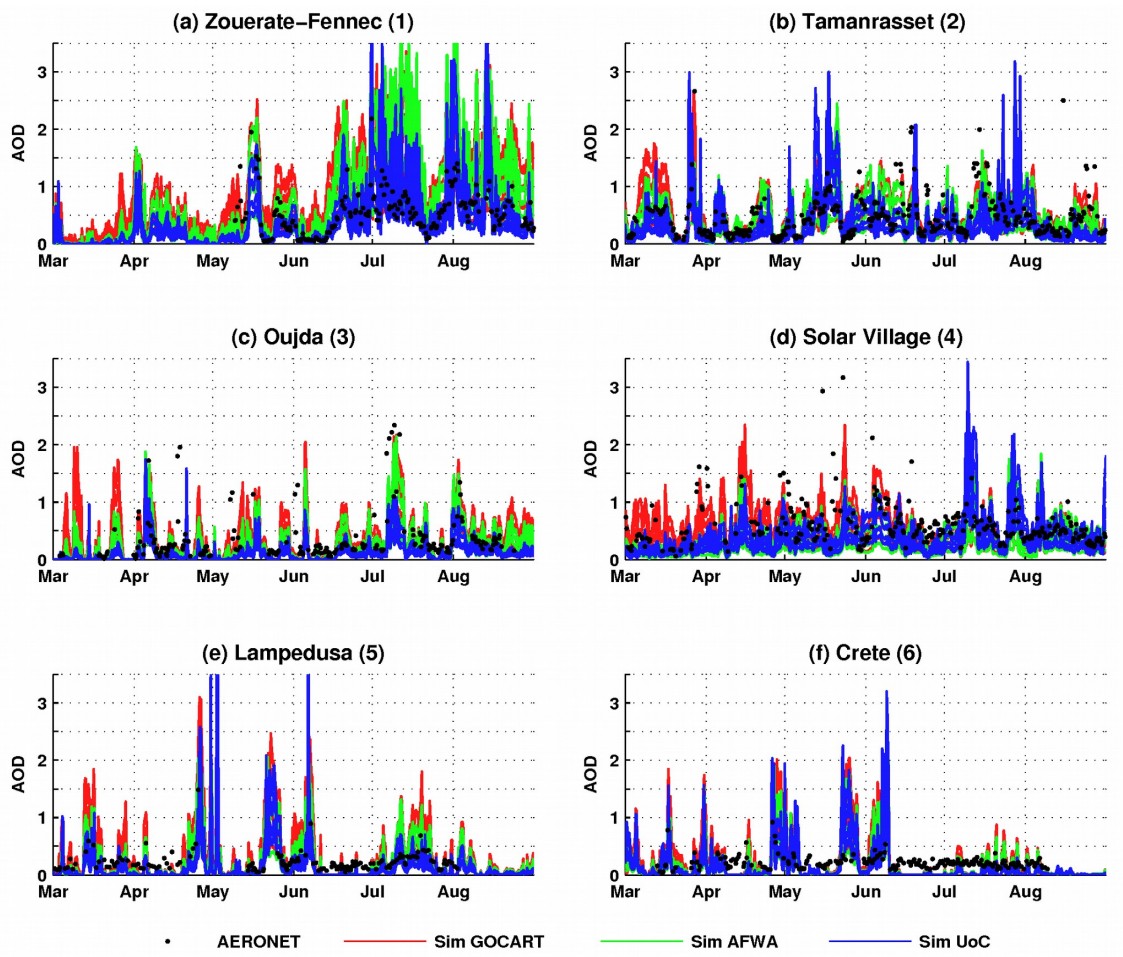

**Figure 9:** Time series of AOD for the simulations and AERONET observations during the whole six-month period. AERONET station locations are shown in Fig. 1.









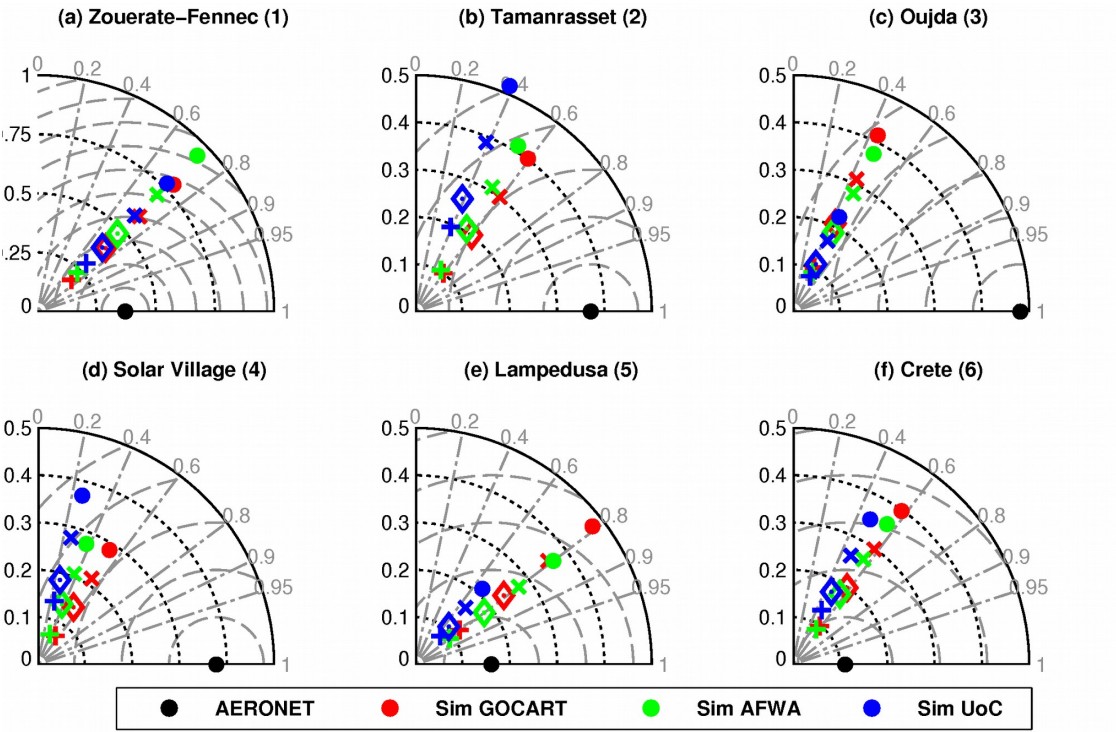

**Figure 10:** Taylor diagram comparing time series of AOD for all simulations to the AERONET station observations as shown in Fig. 9. Root mean square error lines are plotted with a 0.1 interval. Symbol annotations are the same as in Fig. 4.







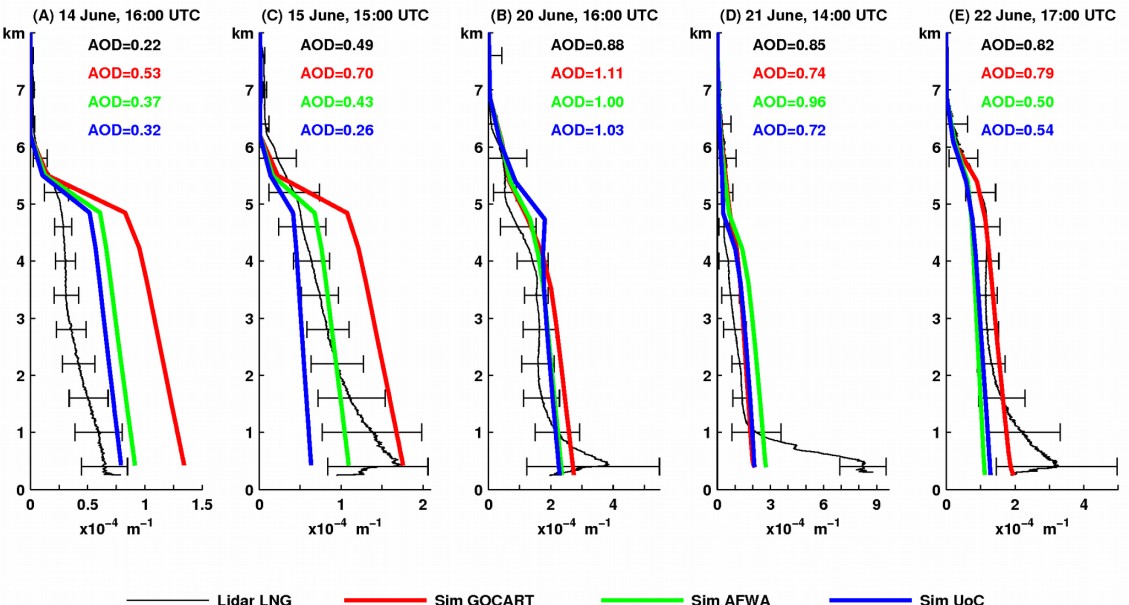

**Figure 11:** Extinction coefficient vertical profiles from the airborne lidar observations (black solid line) and from the WRF-Chem simulations (see legend for colors). The AOD values corresponding to the profiles are shown within the five panels. Error bar lengths equal twice the standard deviations of the lidar measurements at a given altitude.





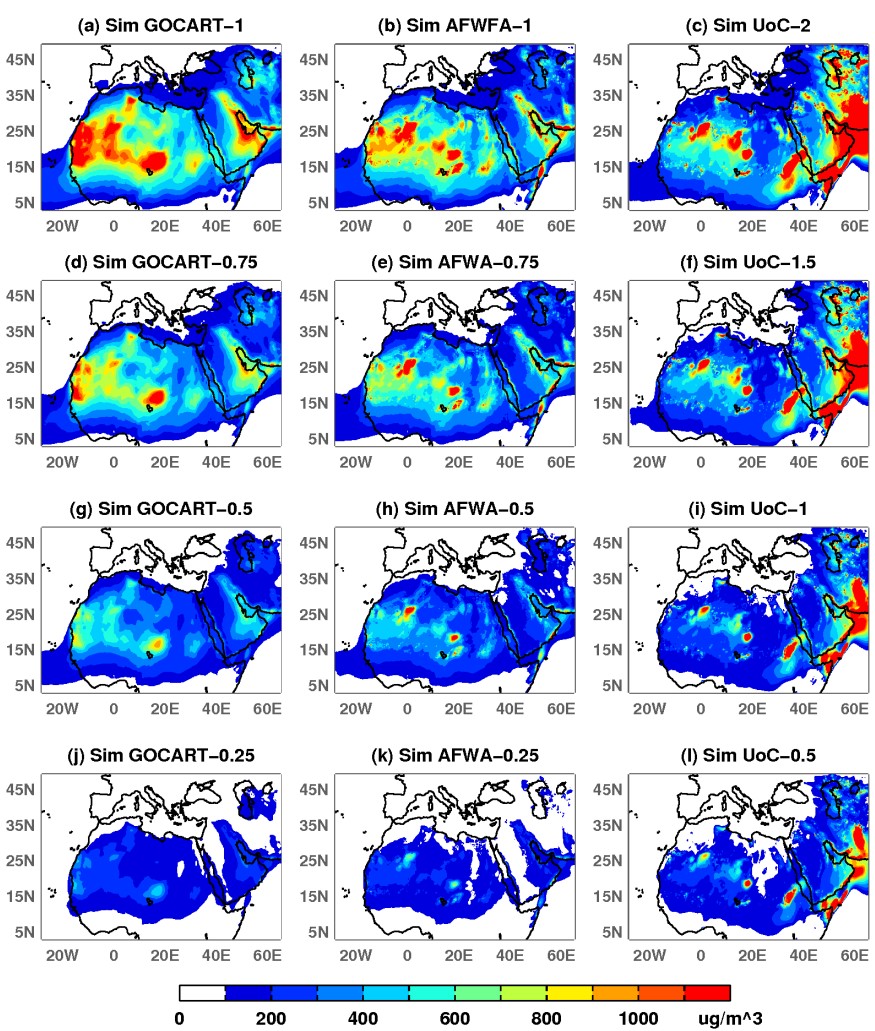

**Figure 12:** Near ground dust concentrations for all simulations, averaged over the 6-month period.



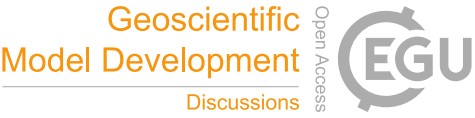



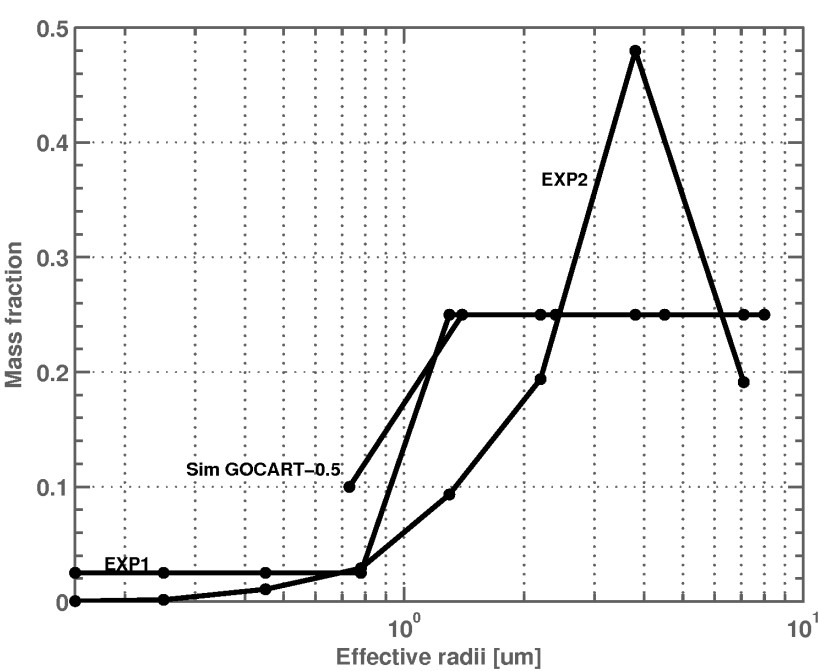

**Figure 13:** Dust bins mass fraction for EXP1, EXP2 and Sim_GOCART-0.5.







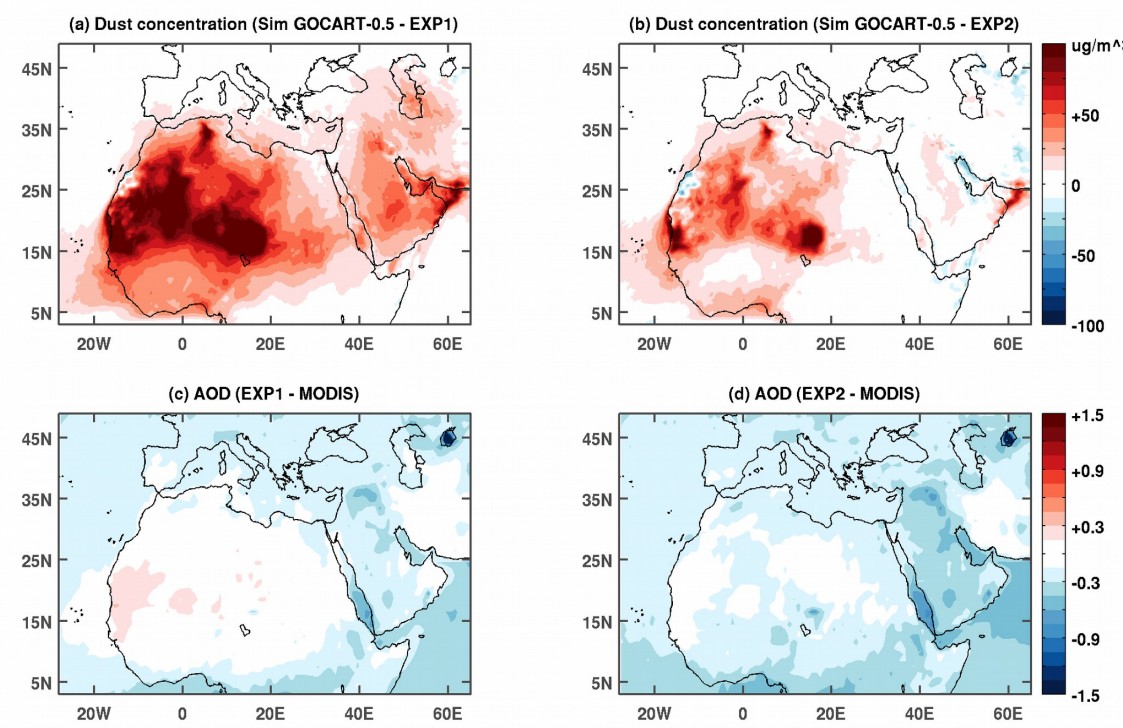

**Figure 14:** A Simulated near ground dust concentration differencess between EXP1 and Sim_GOCART-0.5, averaged over the 6-month period. B as in A but for EXP2. C AOD differences between EXP1 and MODIS observations, averaged over the 6-month period. D as in C but for EXP2.