# Peer review of "Sensitivity of the WRF-Chem (V3.6.1) model to different dust emission parametrisation: Assessment in the broader Mediterranean region"

_Geoscientific Model Development, 2016_

## Referee Comment (RC1) · Anonymous Referee #1 · 7 Mar 2017

General Comments:

The authors evaluated the performance and biases of three different dust emission schemes (GOCART, AFWA, and UoC) available in WRF-Chem. For each scheme, they conducted four different experiments by multiplying various coefficients to the dust emission flux. They also conducted two additional sensitivity experiments, one adding finer dust-size bins and the other changing the mass fraction of each bin, using the GOCART scheme. For each experiment the model was integrated for six months and during the integration the simulation was nudged toward reanalysis. Dust was treated

as passive tracer and thus all simulations use the same meteorological conditions. Model results were evaluated through comparison with observations from MODIS AOD, AERONET AOD, and airborne lidar-derived extinction coefficients. Their evaluation focused on three regions: North Africa and the Arabian Peninsula for dust emission and the eastern Mediterranean for dust transport.

The results show that compared to observations all three schemes perform differently with different multiplication coefficients over dust source regions versus over ocean after transport. However, for the same dust emission scheme, simulations with different multiplication coefficients have similar correlation coefficients with MODIS observations. Results at the simulation domain, regional scale, and local scale (vertical profiles) were also evaluated. They concluded that among the three schemes evaluated none is optimal. However, the multiplication coefficient of 0.5 gave the most reasonable trade-off option between model AOD at both the source regions and transport regions. This work is interesting and the results can be useful to dust modeling and forecasts. The control of the same meteorological conditions, with the use of FDDA, is a good strategy for the evaluation of the dust schemes. The manuscript could be published in GMD after a major revision.

Specific comments:

1. For each dust emission scheme, the differences in performance over the three regions evaluated may not only be due to the scheme itself (i.e., mass fraction, sizes, etc.) but also the quality of observations that are used for comparison. There are common patterns of biases over land versus over ocean and over desert versus over non-desert regions. These systematic biases over different regions might be related to the quality or bias of observations, in addition to the emission schemes themselves. Thus, the authors need to compare observations from different sources and evaluate AOD retrievals from different algorithms (i.e., the deep blue versus dark target algorithms) using another observation (e.g., AERONET). Then observation analyses should be used to help interpret model results with different multiplication coefficients.

2. Sedimentation and wet scavenging are other potential factors that can impact model performance, in particular for the evaluation of dust transport results. The former seems to be included but not the latter. The wet scavenging should be included in model simulations since it has an impact on long-range dust transport.

3. Does the inclusion of additional finer dust-size bins improve the background values (AOD=0.2)?

4. Figure 14a shows that the dust concentration using GOCART-0.5 has a higher value than that in EXP1 near ground. Since EXP1 includes finer bins (thus smaller sedimentation), one would expect to have more dust suspended in the air. If that is the case then results at a higher level, where more dust is expected in EXP1, should be presented as well.

Technical corrections:

1. Line 415: "all models seem to capture..." should read "all experiments seem to capture . . ." since there is only one model (WRF-Chem) used.

2. Line 522: Delete "13" in front of "each".

3. Line 523: Should 0.25 be 0.225?

4. Caption 14 needs attention. (should be (a), (b), . . ., instead of A, B, . . .)

---

## Referee Comment (RC2) · Anonymous Referee #2 · 7 Mar 2017

General comments

This article presents an assessment of the ability of the three-dimensional WRF-Chem model to simulate the transport of dust over the Mediterranean, for a set of dust parameterizations, and over several periods of spring and summer 2011. Model output data are evaluated in comparison with AOD measurements derived from satellite observations, ground-based AERONET stations and airborne lidar-derived extinction coefficient measurements. They focus on the main source area (North Africa, the Arabian Peninsula) and on the Eastern Mediterranean basin. The impact - on this comparison

[Figure]

- of the use of dust emission adjustment coefficients is also investigated.

This topic is of major importance in the Mediterranean, an area which shores are highly populated, which is sensitive to climate change (partly due to atmospheric aerosols), and which is exposed to air quality degradation due to the recurring import of gaseous and particulate pollutants from the surrounding continents.

The model has previously been shown to correctly reproduce meteorological features. The work is of quite good scientific quality, and fits the GMD topics as it proposes a critical analysis of 3D dust emission and transport modelling and aims at the determination of an adequate model set-up.

The questions that arise are the following :

How is the erodibility value obtained? Does the use of a dust flux coefficient aim at scaling this value to better represent dust release during ad hoc wind conditions? Or does it aim at correcting dust emission parameterizations?

§3.1 The authors largely describe the impact of dust flux coefficients on the model skills, in terms of under- or over- estimation. But the analysis of the results remains largely descriptive and not comprehensive. How do the authors explain the spatial heterogeneity in the skills when using the coefficients? Does it come from non-homogeneous quality in the erodibility field above the different areas? Or could it be explained by local soil features that are not all taken into account in the parameterizations? May this come from non-homogeneous local meteorological skills (wind speed restitution)? This issue is only slightly discussed in the conclusion.

§3.4 It does not appear completely satisfactory that the evaluation of the model on the vertical is made using the simulations with the "best local" dust flux coefficients, which are not the same for all parts of the simulation domain. At least, the evaluation of relevance of the model output should thus be limited to the qualitative aspects (restitution of vertical shapes...), and not quantitative ones such as the restitution of the "magnitude

of the extinction coefficients derived from lidar" (line 437).

Technical comments

Line 214 - Reference for AERONET should be given at the first mention of the network and not lately.

Line 522 - "13" should be removed from the sentence.
* * *

---

## Author Comment (AC1) · 31 May 2017

General Comments: The authors evaluated the performance and biases of three different dust emission schemes (GOCART, AFWA, and UoC) available in WRF-Chem. For each scheme, they conducted four different experiments by multiplying various coefficients to the dust emission flux. They also conducted two additional sensitivity experiments, one adding finer dust-size bins and the other changing the mass fraction of each bin, using the GOCART scheme. For each experiment the model was integrated for six months and during the integration the simulation was nudged toward reanalysis.

Dust was treated as passive tracer and thus all simulations use the same meteorological conditions. Model results were evaluated through comparison with observations from MODIS AOD, AERONET AOD, and airborne lidar-derived extinction coefficients. Their evaluation focused on three regions: North Africa and the Arabian Peninsula for dust emission and the eastern Mediterranean for dust transport.

The results show that compared to observations all three schemes perform differently with different multiplication coefficients over dust source regions versus over ocean after transport. However, for the same dust emission scheme, simulations with different multiplication coefficients have similar correlation coefficients with MODIS observations. Results at the simulation domain, regional scale, and local scale (vertical profiles) were also evaluated. They concluded that among the three schemes evaluated none is optimal. However, the multiplication coefficient of 0.5 gave the most reasonable trade-off option between model AOD at both the source regions and transport regions. This work is interesting and the results can be useful to dust modeling and forecasts. The control of the same meteorological conditions, with the use of FDDA, is a good strategy for the evaluation of the dust schemes. The manuscript could be published in GMD after a major revision.

- We would like to thank the Reviewer for carefully reading our manuscript and for his/her positive review. Please note that in this revised version of the article, all simulations have been redone calculating wet removal by large scale precipitation (in addition to convective rainfall removal, activated in the original submission). Finally, we corrected a minor mistake to the calculation of the vertical profiles of extinction coefficient in Fig. 11 that did not significantly affect our results. The text has been adapted to these changes.

Specific comments:

1. For each dust emission scheme, the differences in performance over the three regions evaluated may not only be due to the scheme itself (i.e., mass fraction, sizes, etc.)

[Figure]

but also the quality of observations that are used for comparison. There are common patterns of biases over land versus over ocean and over desert versus over non-desert regions. These systematic biases over different regions might be related to the quality or bias of observations, in addition to the emission schemes themselves. Thus, the authors need to compare observations from different sources and evaluate AOD retrievals from different algorithms (i.e., the deep blue versus dark target algorithms) using another observation (e.g., AERONET). Then observation analyses should be used to help interpret model results with different multiplication coefficients.

- Thank you for pointing this out. We have investigated the quality or bias of MODIS observations in the different regions of interest.

First, we would like to point out that the Deep Blue AOD retrievals (obtained over bright arid land surfaces, such as deserts) are meant to complement the existing Dark Target land and ocean retrievals. Henceforth, there is generally very little geographical overlap between the two retrievals and a comparison between the Deep Blue and Dark Target products is difficult to undertake, and definitely beyond the scope of the present study.

We then have compared AERONET and MODIS Deep blue AODs over Africa (3 stations; Zouerate, Tamanrasset and Oujda) and the Arabian Peninsula (1 station, Solar Village) over the six months of the simulations. We also have compared AERONET and MODIS Dark Target over the Mediterranean for the same period (2 stations: Lampedusa and Crete). The comparisons are shown in Figure A1, together with some relevant indicators such as the absolute bias, root mean square error (RMSE) and correlation. The 1 to 1 line is also shown. Generally speaking, we find very good agreement between AODs in the Mediterranean region (i.e. Dark Target Vs AERONET) with high correlations (0.84 or higher), low RMSE (0.05) and low bias (0.04). On the other hand, the comparison between Deep Blue and AERONET AODs exhibits correlations ranging from 0.39 (Oujda) and 0.83 (Tamanrasset), RMSEs between 0.26 (Zouerate) and 0.55 (Oujda) and biases between 0.19 (Zouerate) and 0.26 (Oujda). These numbers consistently indicate better agreement between Dark Target AODs and AERONET AODs.

Hence, as pointed out by the referee, it is likely that the systematic biases over different regions might be related to the quality or bias of observations, in addition to the emission schemes themselves. It turns out that over Africa and the Middle-East a non-negligible part of the systematic bias observed between MODIS and the WRF simulations may be due to poorer quality of Deep Blue retrievals with respect to the Dark Target ones when compared to the standardized AERONET retrievals. Consistently with our analysis, the additional systematic bias linked to the use of Deep Blue products may be on the order of ∼0.18 (on average for the 4 stations in Africa and in the Middle-East). This is now discussed in the text:

In section 2.3:

"The quality of MODIS observations has been investigated in the different regions of interest. For this reason, MODIS and AERONET observations were compared for the whole six month period of the simulations (March-August, 2011). Deep blue AODs have been evaluated over Africa and the Arabian Peninsula using the four AERONET stations of Zouerate, Tamanrasset, Oujda and Solar Village, while MODIS Dark Target AODs have been evaluated over the Mediterranean using the two AERONET stations of Lampedusa and Crete. Results showed a good agreement between AODs in the Mediterranean region (i.e. Dark Target vs AERONET) with high correlations (0.84 for Crete and 0.95 for Lampedusa), low root mean square errors (RMSEs; 0.05 for both stations) and low absolute bias (0.04 for both stations). On the other hand, the comparison between Deep Blue and AERONET AODs exhibits correlations ranging from 0.39 (Oujda) and 0.83 (Tamanrasset), RMSEs between 0.26 (Zouerate) and 0.55 (Oujda) and biases between 0.19 (Zouerate) and 0.26 (Oujda). These numbers indicate better agreement between Dark Target AODs and AERONET AODs. Consistently with our analysis, the additional systematic bias of AOD linked to the use of Deep Blue products may be on the order of ∼0.18 (on average for the four stations in Africa and in the Middle-East)."

In section 3.1:

"In addition to the emission schemes themselves, the model bias over the dust source regions might be also related to the quality of observations, where MODIS uncertainties over North Africa and Middle-East might be of the order of ~0.18 (section 2.3)."

2. Sedimentation and wet scavenging are other potential factors that can impact model performance, in particular for the evaluation of dust transport results. The former seems to be included but not the latter. The wet scavenging should be included in model simulations since it has an impact on long-range dust transport.

- In the original submission, all simulations were performed with wet removal for convective rain (cumulus scheme). We missed to include this information in the original version of the paper. In this revised version of the paper, simulations and figures have been redone also including wet removal from large scale precipitation (rain due to microphysics scheme).

We now mention that the model treats wet removal explicitly, however the impact of including aerosol removal due to large-scale precipitation on our results was rather insignificant.

3. Does the inclusion of additional finer dust-size bins improve the background values (AOD=0.2)?

- This is an insightful comment. In Figure A2, we present the comparison of EXP1 (red line) and EXP2 (green line) to AERONET observations (similar to Fig. 9 of the article). The model skill in reproducing the background values of AOD does not improve in EXP1 and EXP2. This is clear during June in the AERONET station, located into Crete (panel f).

4. Figure 14a shows that the dust concentration using GOCART-0.5 has a higher value than that in EXP1 near ground. Since EXP1 includes finer bins (thus smaller sedimentation), one would expect to have more dust suspended in the air. If that is the case then results at a higher level, where more dust is expected in EXP1, should be

presented as well.

- Thank you for this comment. Indeed, the vertical profile of dust concentrations in EXP1 and EXP2 are different from GOCART-0.5. We addressed this issue in Fig. 11, where we now include both EXP1 and EXP2. Indeed, the vertical profiles of the two simulations show that dust might reach higher altitudes. We added the following in section 4.2:

"....However, when comparing the vertical profiles of extinction coefficient between the simulations EXP1 and EXP2 with the other simulations (Fig. 11), differences are observed at higher altitudes. For instance, in Fig. 11a the sharp decrease of dust concentration in EXP1 and EXP2 takes place at around 5.5 km with respect to 5 km in the other simulations. In fact, the addition of finer dust sizes suggests a lower rate of sedimentation and therefore differences to the in-column transport of dust. ..."

Technical corrections: 1. Line 415: "all models seem to capture..." should read "all experiments seem to capture . . ." since there is only one model (WRF-Chem) used.

- Done.

2. Line 522: Delete "13" in front of "each".

- Done.

3. Line 523: Should 0.25 be 0.225?

- According to Ginoux et al. (2001), the total size fraction should be equal to 1.1. The total of clay particles should be equal to 0.1 (i.e. 0.025 for each of the four first bins) and equal to 1 for silt (i.e. 0.25 for each of the four last bins).

4. Caption 14 needs attention. (should be (a), (b), . . ., instead of A, B, . . .)

- Caption is now corrected

[Figure]

Figure A1 : Scatter plot of MODIS AODs Vs AERONET AODs for (a) Zouerate, Mauritania, (b) Tamanrasset, Algeria, (c) Oujda, Morocco, (d) Solar Village, Saudi Arabia, (e) Lampedusa, Italy and (f) Crete, Greece. The think black line is the 1-to-1 line, while the thick line shows the best linear regression. Absolute bias, RMSE and correclation coefficients (r²) are also indicated in each panel.

**Fig. 1.**

![Time series plots of AOD for six AERONET sites]

Figure A2: Time series of AOD for the EXP1 (red) and EXP2 (green) simulations and AERONET (black) observations during the whole six-month period.

**Fig. 2.**

---

## Author Comment (AC2) · 31 May 2017

General comments This article presents an assessment of the ability of the three-dimensional WRF-Chem model to simulate the transport of dust over the Mediterranean, for a set of dust parameterizations, and over several periods of spring and summer 2011. Model output data are evaluated in comparison with AOD measurements derived from satellite observations, ground-based AERONET stations and airborne lidar-derived extinction coefficient measurements. They focus on the main source area (North Africa, the Arabian Peninsula) and on the Eastern Mediterranean basin. The

impact -on this comparison- of the use of dust emission adjustment coefficients is also investigated. This topic is of major importance in the Mediterranean, an area which shores are highly populated, which is sensitive to climate change (partly due to atmospheric aerosols), and which is exposed to air quality degradation due to the recurring import of gaseous and particulate pollutants from the surrounding continents.

The model has previously been shown to correctly reproduce meteorological features. The work is of quite good scientific quality, and fits the GMD topics as it proposes a critical analysis of 3D dust emission and transport modelling and aims at the determination of an adequate model set-up.

- We would like to thank the Reviewer for his remarks and for this careful review. Please note that in this revised version of the article we performed new simulations, including wet removal of dust due to large scale precipitation (in addition to wet removal due to convective rainfall, used in the original submission of the article). The Figures have been redone, however the results were only slightly affected. Finally, we corrected a minor mistake to the calculation of the vertical profiles of extinction coefficient in Fig. 11 that did not significantly affect our results.

- The questions that arise are the following: How is the erodibility value obtained? Does the use of a dust flux coefficient aim at scaling this value to better represent dust release during ad hoc wind conditions? Or does it aim at correcting dust emission parameterizations?

- The erodibility field is defined in Ginoux et al, 2001 as a probability field of the areas having accumulated sediments. This is a constant input field to the model, available in a $1°\times1°$ grid. The tuning coefficients that we use in this paper aim at adjusting the modeled dust emissions to a realistic level during ad hoc wind conditions which are identical to all simulations. In fact, only for GOCART and AFWA schemes, where emissions are scaled by the erodibility field, the application of a 0.5 coefficient could be interpreted as a uniform decrease of the erodibility field values by 50%. The relation

of the erodibility field to the tuning coefficient is now added to the last paragraph of section 2.2:

"For each dust emission scheme, we perform four simulations where the dust emissions are multiplied by four different coefficients in order to increase or decrease the dust fluxes in the atmosphere. The erodibility field is used by the GOCART and the AFWA schemes as a scaling factor to dust emissions, meaning that emissions - parametrised as a function of atmospheric and soil physical properties- are scaled in each grid point with different values between 0 and 1. For these schemes the application of a tuning coefficient could be interpreted as a uniform decrease or increase of the erodibility field.Âă More generally, the tuning coefficients applied here aim at scaling the modeled dust emissions to be more realistic and would ideally -for all three schemes- compensate for any boundary conditions or processes that affect dust emission, but are not accounted for in the model. Preliminary tests showed that a coefficient equal to 1 for AFWA and GOCART resulted in disproportionally high AOD values over North Africa compared to the scheme of UoC. Consequently, we chose coefficients to be different for the four simulations when using the UoC scheme. Table 1 presents a summary of the 12 performed simulations set-up."

- §3.1 The authors largely describe the impact of dust flux coefficients on the model skills, in terms of under- or over- estimation. But the analysis of the results remains largely descriptive and not comprehensive. How do the authors explain the spatial heterogeneity in the skills when using the coefficients? Does it come from non-homogeneous quality in the erodibility field above the different areas? Or could it be explained by local soil features that are not all taken into account in the parameterizations? May this come from non-homogeneous local meteorological skills (wind speed restitution)? This issue is only slightly discussed in the conclusion.

- Thank you for this question. The spatial heterogeneity in model skill is likely dependent on the (spatially and temporally) variably accuracy of model input to the schemes, in particular for soil type and vegetation cover, as well as aspects that are not included
or sub-grid in the model, such as information about surface crusting and meteorological processes like dry and moist convection. The tuning constants applied here are only able to provide some insight into the amount of correction needed to compensate for systematic biases and how this amount varies when focusing on particular areas. A conclusive attribution of biases to one or more of the listed sources of error is, however, not possible within the framework of this paper. We have added a paragraph in section 3.1 discussing this issue:

"Figure 3 shows that the average spatial AOD patterns produced by the AFWA and GOCART schemes are similar and differ from those obtained using the UoC scheme. A main reason for this is the scaling of the calculated dust emission fluxes with estimated values for surface erodibility in the AFWA and GOCART implementations. Such empirical tuning is common and necessary in particular for (semi-)empirical parameterizations that do not explicitly describe the physical processes of dust emission at the surface. Physics-based parameterizations, such as the UoC implementation, aim to represent the physics of dust emission and would, if all processes were accounted for, not need empirical tuning. However, dust emission is a complex process including aspects that are not yet accounted for in the parameterizations because they are not yet fully understood and because model resolution limits the spatial representation of land-surface properties. Such aspects include, but are not limited to, surface crusting, particle supply, and intermittency. The spatial variability of model performance, both with and without tuning with constant coefficients, can thus likely be attributed to spatially and temporally varying accuracy of the model lower boundary conditions that are either constant, e.g. soil type, or follow a climatological cycle, e.g. vegetation cover. Surface crusting significantly affects dust emissions, but is to date not represented in any model. Meteorological processes that occur on sub-grid scales in the model, e.g. dry and moist convection, provide another source of uncertainty that can lead to model-observation biases. A conclusive determination of the origins of model over- and underestimations of AOD for the different areas is beyond the scope of this paper. However, an assessment of the tuning required for a particular parameterization

to produce reasonable results can help to determine reasons for model-observation discrepancies."

- §3.4 It does not appear completely satisfactory that the evaluation of the model on the vertical is made using the simulations with the "best local" dust flux coefficients, which are not the same for all parts of the simulation domain. At least, the evaluation of relevance of the model output should thus be limited to the qualitative aspects (restitution of vertical shapes...), and not quantitative ones such as the restitution of the "magnitude of the extinction coefficients derived from lidar" (line 437).

- We agree with the Reviewer that quantifying the model bias in five vertical extinction coefficient profiles is not representative of the model performance. Our purpose here is to provide an insight into the model capacity to realistically reproduce the vertical profile of dust concentration as well as into the uncertainties when comparing model AOD to observations. Quantifying AOD in Fig. 11 we show that even if the model fails to represent the vertical variability of dust concentration, it might reproduce the AOD due to compensating biases. The quantification of AOD only aims to make this point. In section 3.4, we are now more precise on our motivation for presenting Fig. 11:

"Comparing model to observations in only five cases may not be enough to be used as a token of the model performance. On the other hand, Fig. 11 offers an insight into the model capacity to realistically reproduce the vertical variability of dust concentration."

In addition, we slightly changed the concluding remark of section 4.1 to:

"Here we presented only five profiles of extinction coefficient, but averaged over several hundreds of kilometres along the flight legs to average observed outliers, which are thought to be fairly indicative of the model capacity in reproducing vertical profiles of dust concentration."

Technical comments - Line 214 - Reference for AERONET should be given at the first mention of the network and not lately.

- Done.

Line 522 - "13" should be removed from the sentence.

- Done.